

# Bioinformatic analysis and identification of potential prognostic microRNAs and mRNAs in thyroid cancer

Jianing Tang[1], Deguang Kong[2], Qiuxia Cui[1], Kun Wang[3], Dan Zhang[3], Qianqian Yuan[1], Xing Liao[1], Yan Gong[4] and Gaosong Wu[1]

[1] Department of Thyroid and Breast Surgery, Zhongnan Hospital of Wuhan University, Wuhan, China
[2] Department of General Surgery, Zhongnan Hospital of Wuhan University, Wuhan, China
[3] Department of Thyroid and Breast Surgery, Tongji Hospital of Tongji Medical College, Huazhong University of Science and Technology, Wuhan, China
[4] Department of Biological Repositories, Zhongnan Hospital of Wuhan University, Wuhan, China

## ABSTRACT

Thyroid cancer is one of the most common endocrine malignancies. Multiple evidences revealed that a large number of microRNAs and mRNAs were abnormally expressed in thyroid cancer tissues. These microRNAs and mRNAs play important roles in tumorigenesis. In the present study, we identified 72 microRNAs and 1,766 mRNAs differentially expressed between thyroid cancer tissues and normal thyroid tissues and evaluated their prognostic values using Kaplan-Meier survival curves by log-rank test. Seven microRNAs (miR-146b, miR-184, miR-767, miR-6730, miR-6860, miR-196a-2 and miR-509-3) were associated with the overall survival. Among them, three microRNAs were linked with six differentially expressed mRNAs (miR-767 was predicted to target COL10A1, PLAG1 and PPP1R1C; miR-146b was predicted to target MMP16; miR-196a-2 was predicted to target SYT9). To identify the key genes in the protein-protein interaction network , we screened out the top 10 hub genes (NPY, NMU, KNG1, LPAR5, CCR3, SST, PPY, GABBR2, ADCY8 and SAA1) with higher degrees. Only LPAR5 was associated with the overall survival. Multivariate analysis demonstrated that miR-184, miR-146b, miR-509-3 and LPAR5 were an independent risk factors for prognosis. Our results of the present study identified a series of prognostic microRNAs and mRNAs that have the potential to be the targets for treatment of thyroid cancer.

Corresponding authors
Yan Gong, yan.gong@whu.edu.cn
Gaosong Wu, wugaosongtj@163.com

## INTRODUCTION

Thyroid cancer is a common endocrine malignancy which has increased rapidly worldwide in the past decades. The analysis of the Surveillance, Epidemiology, and End Result cancer registry data between 1975 and 2013 revealed that the incidence of thyroid cancer increased by 211% (*Lim et al., 2017*). The improvements in diagnosis and treatment for thyroid cancer substantially improve prognosis (*Grigsby et al., 2006*; *Lang, Wong & Wan, 2013*). According to the histopathological features, thyroid cancer is classified into three major categories: well-differentiated thyroid cancer (WDTC) which includes papillary thyroid
cancer (PTC) and follicular thyroid cancer (FTC); poorly-differentiated thyroid cancer (PDTC) and anaplastic thyroid cancer (ATC) (*Zarkesh et al., 2017*). While WDTC accounts for the majority part of all cases (*Yu et al., 2013*), and its mortality rates were controlled to less than 10% for 10 years, the mortality rates of PDTC and ATC were reported to be 38–57% and close to 100% respectively (*Smallridge et al., 2012*; *Xu & Ghossein, 2016*).

It has been widely accepted that different expression levels of specific genes are associated with cancer initiation. During the past few years, molecular cytogenetics studies have been used to investigate the molecular mechanisms of thyroid cancer. Multiple genes and cellular pathways were reported to participate in the occurrence and development of thyroid cancer. Alterations in the Ras-Raf-mitogen-activated protein kinase (MAPK) pathway are usually observed in WDTC (*Hou et al., 2007*). BRAF mutations were found in 32.4% of PTC cases and RAS mutations in 20–50% of FTC and 13 % of PTC cases (*Nikiforova et al., 2003*; *Sahpaz et al., 2015*). In the ATC, BRAF and RAS alterations were detected in 29% and 23% cases (*Xu & Ghossein, 2016*). Telomerase reverse transcriptase (TERT) promoter mutations were detected in 10% of PTC and related to clinically aggressive behaviors. In addition, mutations of both TERT promoter and BRAF/RAS have a tendency for co-occurrence (*Landa et al., 2016*; *Song et al., 2016*). TP53 mutation is considered to be a genetic event distinguishing WDTC from ATC. Several studies revealed that TP53 mutations were detected in 59% of ATC cases, compared with in 10% WDTC cases (*Kunstman et al., 2015*; *Sykorova et al., 2015*).

With the development of high-throughput technologies, expression profiling of multiple genes is a useful way to find different expression levels of specific genes between normal and tumor tissues. Altered expression levels usually indicate pathological conditions, and proteins coded by these differentially expressed genes may involve in different molecular pathways, biological process, and cellular behaviors during tumor progression. MicroRNAs are small noncoding RNAs which participate in the post-transcriptional regulation of gene expression (*Chen & Kang, 2015*). They function as negative regulators by binding to the $3'$-untranslated region of candidate mRNAs, and repress the gene expression by inhibiting protein translation or degrading mRNAs (*Wang et al., 2010*). Accumulating evidence demonstrated that microRNAs could function as either oncogenes or tumor suppressors in various types of malignancies and regulate different carcinogenic processes (*Liang, Li & Wang, 2017*). In previous bioinformatic studies, a number of microRNAs and mRNAs were identified as predictors for the prognosis of thyroid cancer (Table S1). Due to the small sample sizes and different detection platforms, some results were controversial. In the current study, we analyzed the public microRNA and mRNA sequencing data from The Cancer Genome Atlas Project (TCGA, https://cancergenome.nih.gov/) and identified the differentially expressed microRNAs and mRNAs between thyroid cancer tissues and matched normal thyroid tissues. We then investigated the prognostic value of these differentially expressed microRNAs. In addition, combined with microRNA-mRNA interaction analysis, we analyzed the functions and pathways and constructed protein-protein interaction (PPI) network of differentially expressed mRNAs to investigate the underlying mechanisms of thyroid cancer occurrence and development.

![PeerJ]

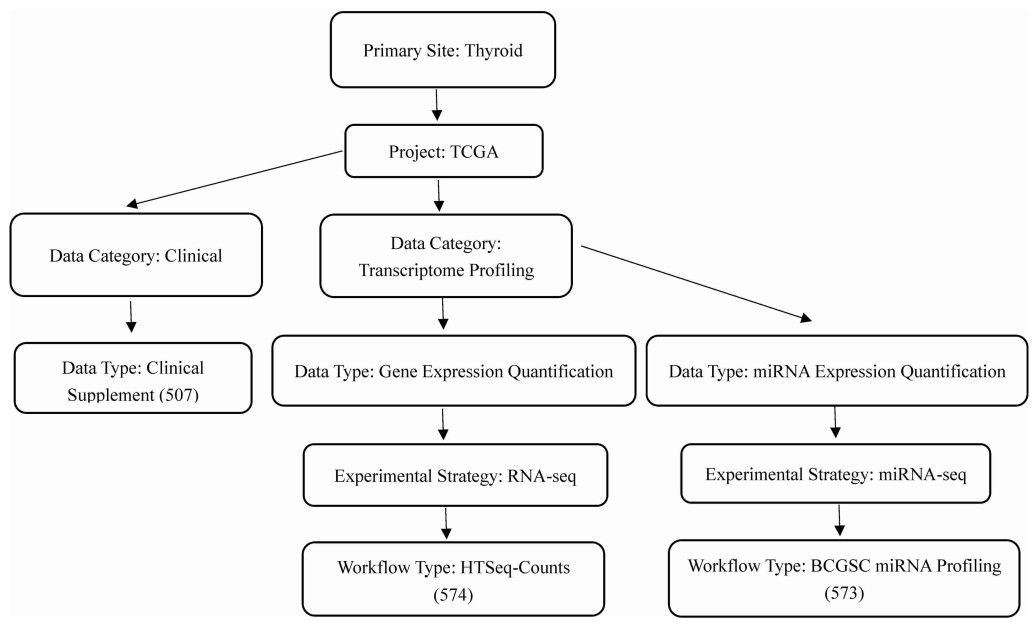

**Figure 1   Workflow of the data selection.**

## MATERIALS AND METHODS

### Data processing

We obtained the clinical information of thyroid and RNA sequencing data from TCGA database (https://cancergenome.nih.gov/). A workflow of the data selection was shown in Fig. 1. Using edgeR package in R language, we normalized the microRNA and mRNA sequencing data and screened out differentially expressed microRNAs and mRNAs between thyroid cancer and normal tissues (*Robinson, McCarthy & Smyth, 2010*). Cut-off criteria of adjusted $p$-value $< 0.01$ and |log2 fold-change (FC)| $> 2$ were considered to be statistically significant.

Clinical information of thyroid cancer patients included sex (female and male), age at diagnosis ($<45$ or $\geq 45$ years), race (White, Black, Asian and other), histological type (classical papillary thyroid cancer, follicular variant of papillary thyroid cancer, tall cell papillary thyroid cancer and other) and tumor-node-metastasis (TNM) stage. Using Kaplan–Meier survival curves by log-rank test, we evaluated the prognostic value of each differentially expressed microRNA with a threshold of $p$-value $< 0.05$. The microRNAs significantly associated with the overall survival were considered as prognostic microRNAs. A Cox proportion hazards model was used to evaluate the relative risk of these prognostic microRNAs on OS. Hazard ratios (HR) with 95% confidence intervals were obtained, any HR $> 1.0$ showed an increased risk of death. A $P$ value $< 0.05$ was considered statistically significant and all tests were two-sided.
## Gene ontology and pathway enrichment analysis of differentially expressed mRNAs

DAVID (http://david.abcc.ncifcrf.gov/) is a database for annotation, visualization and integrated discovery (*Huang da, Sherman & Lempicki, 2009a*; *Huang da, Sherman & Lempicki, 2009b*). Gene Ontology (GO) and KEGG pathway analysis of differentially expressed mRNAs were carried out using DAVID (version 6.8) online tools: functional annotation. The ontology contains three categories: biological process (BP), molecular function (MF) and cellular component (CC). Enriched GO terms and KEGG pathways were identified according to the cut-off criterion of $P$-value $< 0.001$.

## Gene set enrichment analysis

Gene set enrichment analysis (GSEA) is a computational method that determines whether the members of a gene set S are randomly distributed throughout the entire reference gene list L or are found primarily at the top or bottom of L. We performed GSEA using the Java GSEA implementation (supported by Java 8) to validate the enrichment analysis. Annotated gene sets c2.cp.kegg.v6.1.symbols.gmt, c5.bp.v6.1.symbols.gmt, c5.cc.v6.1.symbols.gmt, c5.mf.v6.1.0:0symbols.gmt (Version 6.1 of the Molecular Signatures Database) were chosen as the reference gene sets. FDR $< 0.05$ was set as the cut-off criteria.

## Construction of protein-protein interaction network

In order to investigate the interactive relationships among the differentially expressed mRNAs, we constructed the physical protein-protein interactions of these genes using STRING database (version 10.5) (*Szklarczyk et al., 2017*). PPIs of differentially expressed mRNAs were selected with confidence score $>0.7$. Cytoscape software (http://www.cytoscape.org/) was used to visualize and analyze the PPI network. According to the degree of importance, significant modules of PPI network were screened out using Molecular Complex Detection (MCODE) with the degree cutoff $= 2$, node score cutoff $= 0.2$, $k$-core $= 2$ and max depth $= 100$. CytoHubba was then applied to identify the hub proteins in the PPI network which was widely used explore important nodes in biological networks (*Chin et al., 2014*). Kaplan–Meier survival curves by log-rank test was used to evaluate the prognostic value of each hub gene, a $p$-value $< 0.05$ was set as the cut-of criterion.

## The target genes of prognostic microRNAs

The target genes of prognostic microRNAs were predicted using miRTarBase (version 6.0) (http://mirtarbase.mbc.nctu.edu.tw/php/index.php), miRDB (version 5.0) (http://www.mirdb.org/miRDB/) and TargetScan (version 7.1) (http://www.targetscan.org/) databases. The overlapping target genes were identified to further enhance the reliability of bioinformatics analysis. The overlapping target genes with differentially expressed mRNAs were then compared, and the microRNA-mRNA network was visualized by cytoscape software.
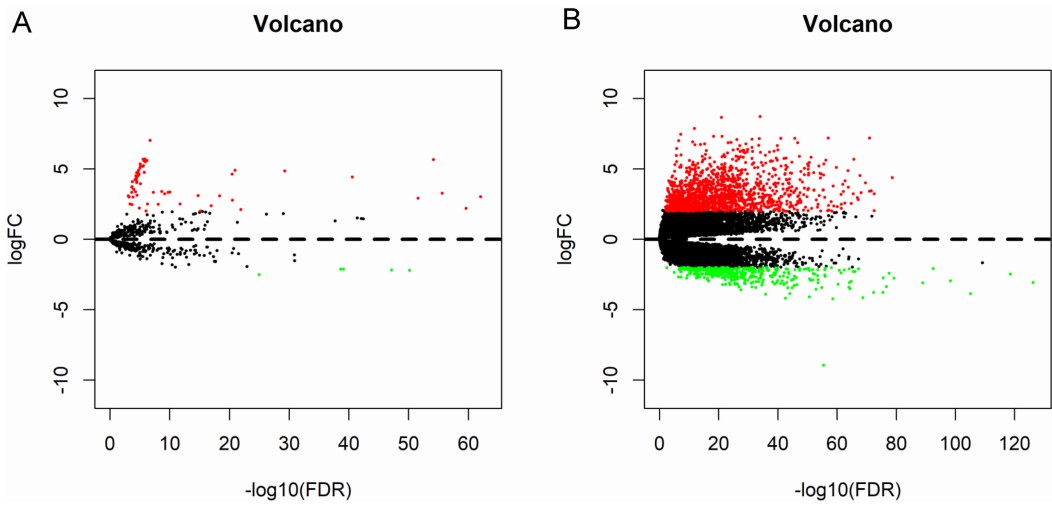

**Figure 2** **Volcano plot of differentially expressed microRNAs and mRNAs.** The red dots represent up-regulated genes, and green dots represent down-regulated genes. (A) microRNA; (B) mRNA.

## RESULTS

### Identification of differentially expressed microRNAs and mRNAs in thyroid cancer

A total of 573 samples with microRNA sequencing data and 568 samples with mRNA sequencing data were analyzed. The microRNA sequencing data included 514 thyroid cancer samples and 59 matched normal samples; and the RNA sequencing data included 510 thyroid cancer samples and 58 matched normal samples. Based the cut-off criteria ($P < 0.01$ and $|\log 2FC| > 2.0$), a total of 72 differentially expressed microRNAs and 1766 differentially expressed mRNAs were identified between thyroid cancer tissues and matched normal tissues, including 67 up-regulated and five down-regulated microRNAs; 1,370 up-regulated and 396 down-regulated genes. The results were presented as Volcano plot (Fig. 2).

### Association between differentially expressed microRNAs and clinical features

To identify the potential prognostic microRNAs correlated with the overall survival of thyroid cancer patients, we investigated the associations between miRNAs expression and patient survival using Kaplan–Meier curve and Log-rank test. Seven microRNAs were related to the overall survival. While five of them (miR-146b, miR-184, miR-767, miR-6730 and miR-6860) were positively correlated with the overall survival, the other two microRNAs (miR-196a-2 and miR-509-3) were negatively related to the overall survival (Fig. 3). In the multivariate Cox analysis, only miR-184, miR-146b and miR-509-3 were significantly associated with overall survival when controlling for the remaining variables: age at diagnosis, sex, stage, tumor size, lymph nodes metastases, distant metastases and other diagnostic genes revealed by Kaplan–Meier curve (Table 1). Interestingly, all of the seven microRNAs were up-regulated in the tumor samples. MiR-146b, miR-509-3 and

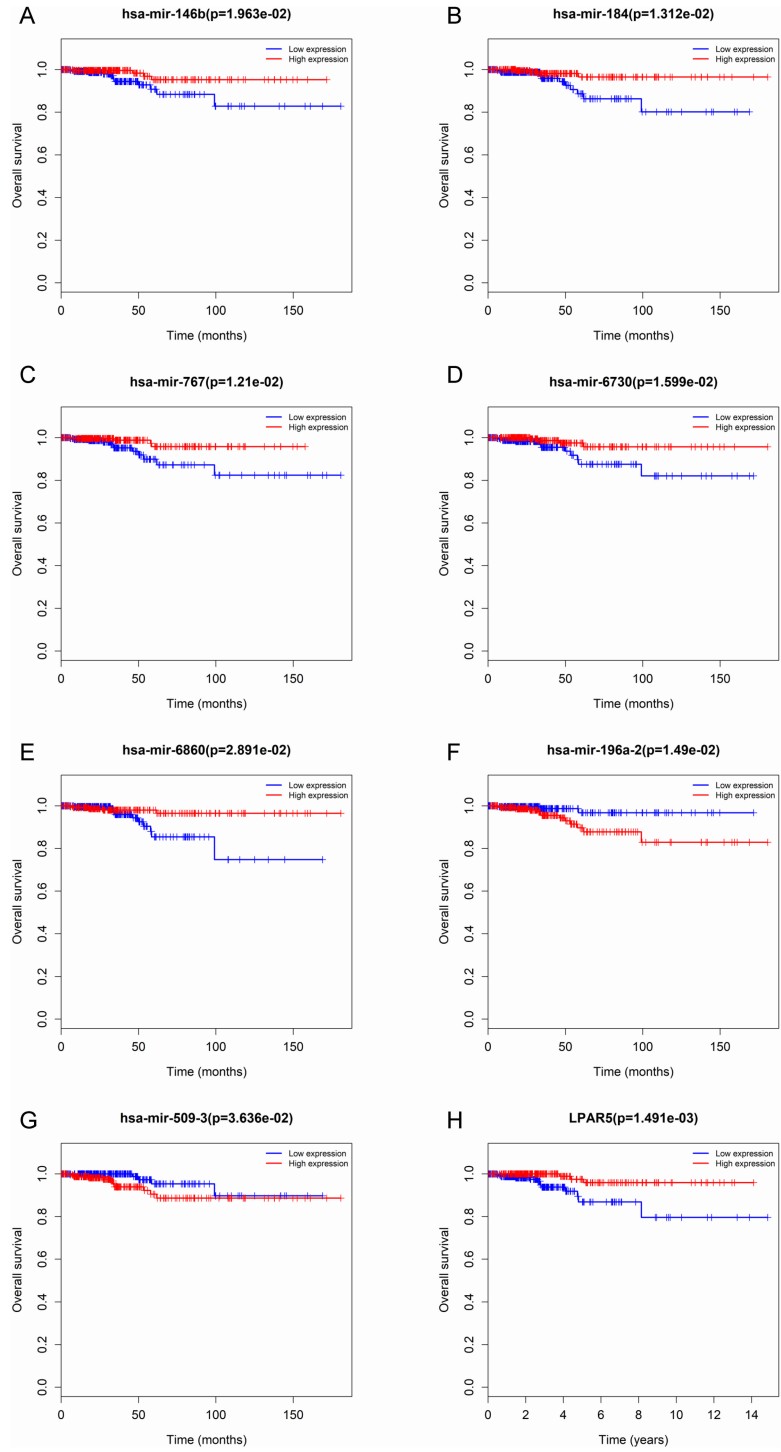

**Figure 3** **Seven microRNAs and 1 mRNA were associated with the overall survival in thyroid cancer patients using Kaplan–Meier survival curves by log-rank test.** The patients were stratified into high-level group and low-level group according to median expression. (A) miR-146b; (B) miR-184; (C) miR-767; (D) miR-6730; (E) miR -6860; (F) miR-196a-2; (G) miR-509-3; and (H) LPAR5.

**Table 1  Cox proportional hazards regression model analysis of factors associated with overall survival.**

| Variable | Multivariate hazard ration | 95% CI | *P-value |
|---|---|---|---|
| mir-767 expression High vs. Low | 0.316 | 0.071–1.412 | 0.131 |
| mir-6730 expression High vs. Low | 0.370 | 0.081–1.685 | 0.199 |
| mir-184 expression High vs. Low | 0.201 | 0.045–0.901 | 0.036 |
| mir-6860 expression High vs. Low | 0.729 | 0.142–2.749 | 0.706 |
| mir-146b expression High vs. Low | 0.251 | 0.065–0.970 | 0.035 |
| mir-196a-2 expression High vs. Low | 2.864 | 0.065–4.881 | 0.147 |
| mir-509-3 expression High vs. Low | 4.534 | 2.281–9.923 | 0.002 |
| LPAR5 expression High vs. Low | 0.043 | 0.012–0.453 | 0.005 |

**Notes.**
*P values calculated by multivariate Cox analysis (adjusted for age at diagnosis, sex, stage, tumor size, lymph nodes metastases and distant metastases).
CL, confidence interval.

miR-6730 were significantly associated with tumor stage and lymph node metastases; only miR-509-3 was related to tumor size (Fig. 4).

## Gene ontology and pathway enrichment analysis

Through GO analysis, the enriched go terms were classified into BP, MF and CC. Our results showed that the most significantly enriched GO terms corresponded to up-regulated genes were extracellular matrix (ECM) organization (BP), calcium ion binding (MF) and proteolysis (CC). While the down-regulated genes were mainly enriched in receptor localization to synapse (BP), heparin binding (MF) and proteinaceous extracellular matrix (CC) (Table 2). According to Kyoto Encyclopedia of Genes and Genomes (KEGG) pathway analysis, up-regulate genes were significantly enriched in neuroactive ligand–receptor interaction, protein digestion and absorption, pancreatic secretion, complement and coagulation cascades, salivary secretion, ECM-receptor interaction, nicotine addiction, cell adhesion molecules and morphine addiction. Down-regulate genes were mainly enriched in complement, coagulation cascades and drug metabolism—cytochrome P450 (Table 3).

## Gene set enrichment analysis

Based on the cut-off criteria FDR < 0.05, only three functional gene sets were enriched: "Notch signaling pathway" "Glycosaminoglycan degradation" and "P53 signaling pathway" (Fig. 5).

## Protein-protein interaction and mircoRNA-target network analysis

To predict protein interactions, the 1,766 differentially expressed genes were submitted to the Search Tool for the Retrieval of Interacting Gene (STRING) database. The PPI network

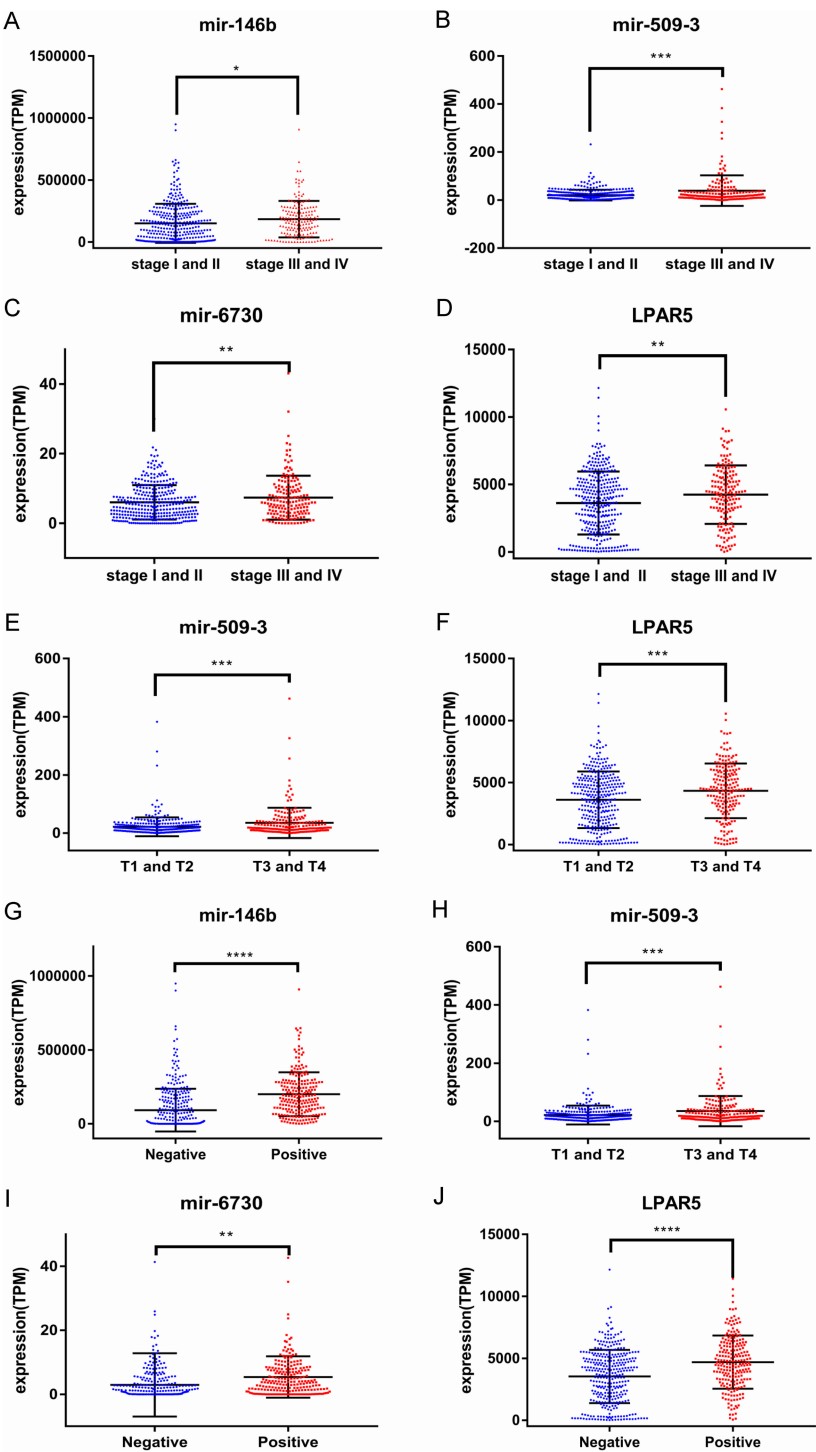

**Figure 4  Correlations between the expression levels of prognostic genes and tumor pathology.** (A) miR-146b and tumor stage; (B) miR-509-3 and tumor stage; (C) miR-6730 and tumor stage; (D) LPAR5 and tumor stage; (E) miR-509-3 and tumor size; (F) LPAR5 and tumor size; (G) miR-146b and lymph nodes metastases; (H) miR-509-3 and lymph nodes metastases; (I) miR-6730 and lymph nodes metastases; (J) LPAR5 and lymph nodes metastases.

**Table 2  The top 5 enriched gene ontology terms of differentially expressed genes.**

| Expression | Category | Term | Gene count | Adjust P value |
|---|---|---|---|---|
| Up-regulated | GOTERM_BP | GO:0030198~extracellular matrix organization | 38 | 1.01E−13 |
| | GOTERM_BP | GO:0030574~collagen catabolic process | 20 | 2.87E−11 |
| | GOTERM_BP | GO:0008544~epidermis development | 22 | 1.23E−10 |
| | GOTERM_BP | GO:0007155~cell adhesion | 52 | 2.77E−09 |
| | GOTERM_BP | GO:0006508~proteolysis | 54 | 6.99E−09 |
| | GOTERM_MF | GO:0005509~calcium ion binding | 75 | 9.53E−12 |
| | GOTERM_MF | GO:0004252~serine-type endopeptidase activity | 38 | 2.02E−10 |
| | GOTERM_MF | GO:0005198~structural molecule activity | 34 | 1.58E−08 |
| | GOTERM_MF | GO:0008201~heparin binding | 26 | 4.01E−08 |
| | GOTERM_MF | GO:0004869~cysteine-type endopeptidase inhibitor activity | 12 | 1.39E−07 |
| | GOTERM_CC | GO:0006508~proteolysis | 209 | 5.72E−46 |
| | GOTERM_CC | GO:0005615~extracellular space | 170 | 4.03E−35 |
| | GOTERM_CC | GO:0005578~proteinaceous extracellular matrix | 44 | 5.86E−13 |
| | GOTERM_CC | GO:0005887~integral component of plasma membrane | 117 | 3.75E−10 |
| | GOTERM_CC | GO:0031012~extracellular matrix | 36 | 2.73E−07 |
| Down-regulated | GOTERM_BP | GO:0097120~receptor localization to synapse | 4 | 1.02E−04 |
| | GOTERM_BP | GO:0035418~protein localization to synapse | 4 | 3.36E−04 |
| | GOTERM_BP | GO:0097114~NMDA glutamate receptor clustering | 3 | 0.001 |
| | GOTERM_BP | GO:0023041~neuronal signal transduction | 3 | 0.002 |
| | GOTERM_BP | GO:0097119~postsynaptic density protein 95 clustering | 3 | 0.002 |
| | GOTERM_MF | GO:0008201~heparin binding | 7 | 0.006 |
| | GOTERM_MF | GO:0005509~calcium ion binding | 16 | 0.007 |
| | GOTERM_CC | GO:0005578~proteinaceous extracellular matrix | 14 | 7.32E−06 |
| | GOTERM_CC | GO:0005615~extracellular space | 34 | 7.77E−06 |
| | GOTERM_CC | GO:0005576~extracellular region | 37 | 2.11E−06 |
| | GOTERM_CC | GO:0030141~secretory granule | 6 | 0.001 |
| | GOTERM_CC | GO:0043025~neuronal cell body | 11 | 0.002 |

**Notes.**
BP, biological process; MF, molecular function; CC, cellular component.
P value < 0.01 was considered as threshold values of significant difference.

consisted of 483 nodes and 1,437 edges. The target genes of prognostic microRNAs were compared with these differentially expressed genes, and six target genes regulated by three microRNAs were identified in the network (miR-767 was predicted to target COL10A1, PLAG1 and PPP1R1C; miR-146b was predicted to target MMP16; miR-196a-2 was predicted to target SYT9) (Fig. S1). Moreover, the network complex was further analyzed, and the most significant module was screened out using MCODE, which contained 26 nodes and 325 edges (Fig. 6). Then we performed KEGG pathway analysis of these 26 genes, our results demonstrated that they were mainly involved in neuroactive ligand–receptor interaction, chemokine signaling pathway and cAMP signaling pathway. We screened out the top 10 hub nodes with higher degrees using the plug-in CytoHubba in Cytoscape. These hub genes included neuropeptide Y (NPY), neuromedin U (NMU), kininogen 1 (KNG1), lysophosphatidic acid receptor 5 (LPAR5), C–C motif chemokine receptor 3

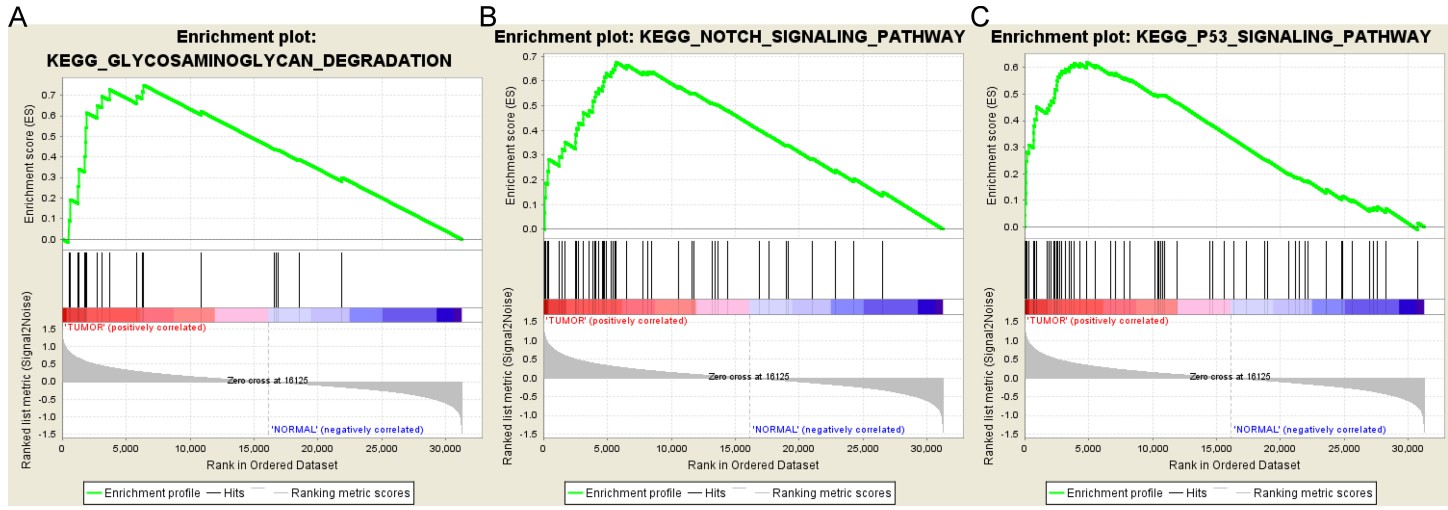

**Figure 5** **Gene set enrichment analysis.** (A) Glycosaminoglycan degradation pathway. (B) Notch signaling pathway. (C) P53 signaling pathyway.

**Table 3** **KEGG pathway analysis of differentially expressed genes.**

| Expression | Pathway | Gene count | Adjust *P* value |
|---|---|---|---|
| Up-regulated | hsa04080: Neuroactive ligand–receptor interaction | 33 | 7.78E−07 |
| | hsa04974: Protein digestion and absorption | 15 | 3.53E−05 |
| | hsa04972: Pancreatic secretion | 15 | 6.62E−05 |
| | hsa04610: Complement and coagulation cascades | 11 | 9.91E−04 |
| | hsa04970: Salivary secretion | 12 | 0.002 |
| | hsa04512: ECM-receptor interaction | 12 | 0.002 |
| | hsa05033: Nicotine addiction | 8 | 0.002 |
| | hsa04514: Cell adhesion molecules (CAMs) | 15 | 0.005 |
| | hsa05032: Morphine addiction | 11 | 0.008 |
| Down-regulated | hsa04610: Complement and coagulation cascades | 6 | 8.62E−04 |
| | hsa00982: Drug metabolism—cytochrome P450 | 5 | 0.006 |

**Notes.**
    *P* value < 0.01 was considered as threshold values of significant difference.

(CCR3), somatostatin (SST), pancreatic polypeptide (PPY), gamma-aminobutyric acid type B receptor subunit 2 (GABBR2), adenylate cyclase 8 (ADCY8) and serum amyloid A1 (SAA1). Survival analysis of the 10 hub genes demonstrated that only LPAR5 was significantly correlated with the overall survival (Fig. 3H). Multivariate Cox analysis demonstrated that LPAR5 was an independent risk factors for overall survival (Table 1). And the expression of LPAR5 was associated with tumor stage, size and lymph node metastases (Fig. 4).

## Validation using GEO database

To validate the results, we screened out the differentially expressed genes using GSE3467 and GSE73182. GSE3467 was an expression microarray dataset and GSE73182 was a miRNA expression microarray dataset. Based on the same cut-off criteria, 130 differentially

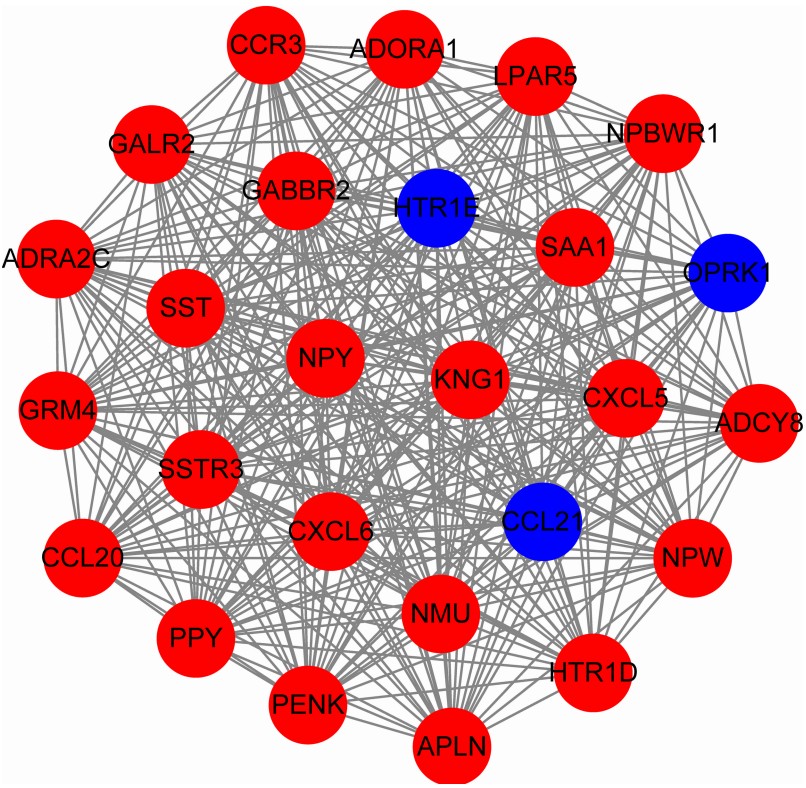

**Figure 6** **The significant module consists of 26 nodes and 325 edges.** The red nodes represent up-regulated genes, and blue nodes represent down-regulated genes.

expressed mRNAs and five differentially expressed miRNAs were identified. The overlapping genes (95 mRNAs and four miRNAs) were shown in Fig. S2.

## DISCUSSION

As one of the most common endocrine malignancies, it is important to investigate the molecular mechanisms of thyroid cancer occurrence and development. In the current study, we investigated potential microRNAs and mRNAs correlated with thyroid tumorigenesis using bioinformatics analysis. A total of 72 differentially expressed microRNAs and 1,766 differentially expressed genes were identified from TCGA database, including 67 up-regulated and five down-regulated microRNAs;1370 up-regulated and 396 down-regulated genes. MiR-146b, miR-184, miR-767, miR-6730, miR-6860, miR-196a-2, miR-509-3 and LPAR5 were correlated with the overall survival of thyroid cancer patients. GSEA analysis demonstrated that the gene sets "Notch signaling pathway" "Glycosaminoglycan degradation" and "P53 signaling pathway" were significantly enriched in thyroid cancer samples.

In the last decade, microRNAs have been revealed to modulate multiple processes of cancer development, including cancer cell proliferation, differentiation, apoptosis, migration and invasion. However, the studies of microRNAs' effects on cancer prognosis

was limited due to the small sample size, different detection platforms and the lack of long-term follow-up. To investigate the potential prognostic microRNAs of thyroid cancer, we analyzed the high-throughput data from TCGA database and identified seven prognostic microRNAs associated with the clinical outcome of thyroid cancer patients. Geraldo et al. stated that miR-146b-5p was highly expressed in papillary thyroid cancer and considered as a relevant diagnostic marker for this type of thyroid cancer. MiR-146b-5p was reported to promote the migration and invasion of papillary thyroid cancer cells via downregulating ZNRF3 and upregulating Wnt/catenin signaling pathway, and to promote thyroid follicular cell growth via downregulating TGF-$\beta$ pathway by binding to the 3′-untranslated region of SMAD4 (*Chou, Liu & Kang, 2017*; *Deng et al., 2015*; *Geraldo, Yamashita & Kimura, 2012*). Inversely, the expression levels of miR-146b was positively related to the overall survival of patients with thyroid cancer. The expression levels of miR-509-3 were lower in multiple cancers, and miR-509-3-5P downregulation promoted the migration and invasion of gastric cancer cells by targeting PODXL. Overexpression of miR-509-5p markedly inhibited the proliferation, migration and invasion of non-small lung cancer cells via targeting YWHAG (*Peng, Yu & Fu, 2016*; *Zhang et al., 2017*). While miR-509-3 was overexpressed in thyroid cancer and negatively related to the overall survival, miR-184 was considered as a potential oncogenic microRNA of squamous cell carcinoma via promoting cancer cell proliferation. Inversely, miR-184 was down-regulated in aggressive human glioma and breast cancer cells, and inhibited cancer cell proliferation and invasion (*Emdad et al., 2015*; *Feng & Dong, 2015*). In the present study, miR-184 was positively related to the overall survival of patients with thyroid cancer. MiR-767 was reported to represses TET1/3 (two tumor suppressor genes) and identified as a hallmark of cancer (*Loriot et al., 2014*). The molecular mechanisms are still to be investigated. Previous studies have identified a number of miRNAs involved in thyroid carcinogenesis (Table S2). MiR-146b, miR-221 and miR-222 were the most three frequent miRNAs reported in thyroid cancer, they appeared to associated with high-risk features such as extrathyroidal extension, lymph node metastasis, distant metastasis and BRAF[V600E] mutation. While in our present study, the results were partly different from the previous: miR-221, miR-222 and some other miRNAs were not identified. This might because of the following reasons: different study types (RT-PCR, array, high throughput sequencing), different platforms, different ways for normalization, different ways for DEG screening, and the cut-off criteria (in our study $P < 0.01$ and $|\log2FC| > 2.0$). In our validation datasets GSE3467 and GSE73182, 130 differentially expressed mRNAs and five differentially expressed miRNAs were identified, among them, 95 differentially expressed mRNAs and 4 differentially expressed miRNAs were overlapping with TCGA dataset.

Through integrated bioinformatics analysis, we identified the most significant module which contained 26 nodes and 325 edges, and these genes were mainly enriched in neuroactive ligand–receptor interaction, chemokine signaling pathway and cAMP signaling pathway. The top 10 hub genes with higher degrees were NPY, NMU, KNG1, LPAR5, CCR3, SST, PPY, GABBR2, ADCY8 and SAA1. NPY encodes a neuropeptide which influences multiple physiological processes through G protein-coupled receptors (GPCRs) and MAPK. NPY was reported to promote inflammation-induced tumorigenesis via PI3-K/Akt

pathway and miR-375-dependent apoptosis (*Jeppsson, Srinivasan & Chandrasekharan, 2016*). NMU was associated with increased breast cancer aggression (*Martinez et al., 2017*) and its overexpression induced regional metastasis of head and neck squamous cell carcinoma (*Wang et al., 2016*). KNG1 was identified as a potential marker of early colorectal cancer stages (*Quesadacalvo et al., 2017*). LPAR5 is a member of the rhodopsin class GPCRs. In our study, it was positively corrected with the overall survival of thyroid cancer patients. LPAR5 was down-regulated in primary undifferentiated nasopharyngeal carcinoma and promoted the LPA-induced migration of nasopharyngeal carcinoma cell lines (*Yap et al., 2015*). Additionally, LPAR5 negatively regulated cell motile and invasive activities of human sarcoma cell lines (*Dong et al., 2014*). CCR3 belongs to family 1 of the GPCRs, which enhance cellular proliferation, invasion, and migration through ERK and JNK signaling pathway (*Dong et al., 2014*). CCRs was reported to correlated with improved distant relapse-free survival in breast cancer (*Gong et al., 2016*). SST is a neuropeptide which affects neurotransmission, secretion and cell proliferation. The receptor of SST was reported to be a predictor of better response to therapy in medullary thyroid carcinoma (*Kendler et al., 2017*). In radiation-induced papillary thyroid cancer from chernobyl pediatric patients, GABBR2 was highly expressed (*Stein et al., 2010*), and related to tumor occurrence. ADCY8 is a membrane bound enzyme which is differentially expressed in endometrial cancer (*Orchel et al., 2012*). SAA1 is highly expressed in response to tissue injury and inflammation, and its highly expression is associated with chronic inflammation, lipid metabolism and tumor pathogenesis (*Sun & Ye, 2016*). SAA1 has been used as a non-invasive biomarker for the prognosis of many cancers, including stomach, breast, liver and lung neoplasms (*Knebel et al., 2017*; *Upur et al., 2015*).

## CONCLUSION

In conclusion, our study identified the crucial microRNAs and mRNAs in thyroid cancer, and constructed the regulatory network between microRNAs and mRNAs through bioinformatics analysis. A total of 72 differentially expressed microRNAs and 1,766 differentially expressed genes were screened out. Among them, seven microRNAs were correlated with the overall survival. Among the hub genes identified from PPI network, LPAR5 may play important roles in thyroid cancer. Multivariate analysis demonstrated that miR-184, miR-146b, miR-509-3 and LPAR5 were an independent risk factors for prognosis, and they have the potential to be the targets for treatment of thyroid cancer. However, further experimental research is still required to confirm the functions of identified genes.

### Funding
The authors received no funding for this work.

### Competing Interests
The authors declare there are no competing interests.

## Author Contributions

- Jianing Tang conceived and designed the experiments, performed the experiments, analyzed the data, prepared figures and/or tables, authored or reviewed drafts of the paper, approved the final draft.
- Deguang Kong, Qiuxia Cui, Dan Zhang and Xing Liao performed the experiments, analyzed the data, prepared figures and/or tables, authored or reviewed drafts of the paper, approved the final draft.
- Kun Wang and Qianqian Yuan performed the experiments, analyzed the data, contributed reagents/materials/analysis tools, prepared figures and/or tables, authored or reviewed drafts of the paper, approved the final draft.
- Yan Gong conceived and designed the experiments, authored or reviewed drafts of the paper, approved the final draft.
- Gaosong Wu conceived and designed the experiments, prepared figures and/or tables, authored or reviewed drafts of the paper, approved the final draft.

## Data Availability

The raw data are provided as Supplemental Files.

## Supplemental Information

Supplemental information for this article can be found online at http://dx.doi.org/10.7717/peerj.4674#supplemental-information.

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
