# Peer review of "Bioinformatic analysis and identification of potential prognostic microRNAs and mRNAs in thyroid cancer"

_PeerJ, doi:10.7717/peerj.4674_

## Round 0.1 · original submission · Major Revisions

Please address all comments from the two reviewers. The second reviewer has raised significant concerns about the use methods. Please pay special attention in addressing these comments.

·

Basic reporting

The manuscript from Jianing Tang, and colleagues identified the differentially expressed microRNAs and mRNAs between cancer tissues and matched normal thyroid tissues by public microRNA and mRNA sequencing data. They also investigated the mechanisms of thyroid cancer by constructed protein-protein interaction (PPI) network of differentially expressed mRNAs combined with microRNA-mRNA interaction analysis. Overall, I think the authors supplied useful candidate genes and theoretical basis of mechanisms for thyroid cancer research.

I have several major concerns about this paper.

1. I agree with specific gene changes are associated with cancer initiation, however, what is the relationship between specific gene changes and gene expression change. I mean, a specific gene changed means its expression level change or its protein production change. If I am not understanding wrong, the DE genes identified in this manuscript between cancer tissue and normal tissue should be the same gene at different expression level in different tissues. I think the introduction needs to elaborate what kind of gene change the authors are looking at and explain more about that.

2. What is the quality of raw microRNAs and mRNAs data used in this paper? I suggest add this information as a supplemental table here.

Minor concerns:
1. Paragraph transitions need to improve in the introduction part.
2. For Gene ontology and pathway enrichment analysis, I am wondering why not use GSEA for trancriptome analysis?
3. Are the pvalues in Table1 and Table2 adjust p value? If yes, please label “adjust P value” in the tables.
4. The resolution of figures need to be improved.

Experimental design

no comment

Validity of the findings

no comment

Additional comments

Generally, I like this work and the idea generated by the authors. The introduction part really needed to improved and more detailed.

·

Basic reporting

In this study, the authors re-analyse TCGA data from thyroid cancer samples in order to find micro RNAs (miRNAs) and mRNAs that can be used as prognostic factors. In order to do this, they do analyses to find differentially expressed miRNAs and mRNAs between cancer and normal samples, and then do GO-term enrichment analyses and build a protein-protein interaction network in order to try and devise biological pathways that might be altered in thyroid cancer. Generally, the Methods section needs substantially more detail to be able to follow what the authors have done, and to put the relevance of their results into context. This could be helped in the introduction by adding some references to previous studies that explore this same issue, and saying how this study differs from these previous ones and ideally how it improves the analysis. I think that figures and tables are well presented, even if hard to understand in a few cases (details below). The discussion needs to include not only results that agree with the authors’ analyses but also those that don’t, and it should be discussed or speculated why this may be the case. The manuscript is written in understandable English, but it can be improved (one example would be changing “…substantially promote prognosis” (lines 37-38) for “substantially improve prognosis”. Ideally, a replication of these results should be attempted if another dataset is available for analysis (maybe in dbGaP?).

Experimental design

In general, the Methods section needs much more detail to be understood and replicated.

In the Data Processing section, it is only mentioned that “clinical information of thyroid and RNA sequencing data from TCGA database” were obtained. No details are offered regarding the filters applied for downloading data, if any (for example, I assume they are talking about TCGA-THCA, thyroid carcinoma? Because there is also some thyroid data in the FM-AD project. Additionally, were all tissue types included? Did they download HTSeq raw counts, FPKM or FPKM-UQ data? Did they download only tumours with matched normal? In the second paragraph of this section authors mention that clinical information for patients included race: White, Black or Asian but there is one patient in the TCGA-THCA dataset with American Indian/Alaska Native ancestry. Was this patient excluded?). Authors also do not give any details about what they used to normalise these data, they just mention it was done “with R language package”. The package used and the initial dataset downloaded can have a huge impact on the results, for example, there are some R packages that already perform normalisation and thus raw HTSeq counts must be used with these. Clarifying these details is necessary before a reader can understand the methods.

In the Kaplan-Meier/log-rank analyses, it is mentioned that a p-value threshold of 0.05 was used to determine the prognostic value of differentially expressed miRNAs, but it is not mentioned if a multiple testing correction was applied to these p-values.

In the Gene Ontology and pathway enrichment analysis, what methods were used to find enriched GO terms? It is mentioned that online tools from the website DAVID were used, but it is not mentioned which ones. What version of the tools were used? Does it use hypergeometric tests to obtain enriched pathways? What identifiers were used?

The same with the protein-protein interaction network: What version of STRING was used? What of the many different datasets were downloaded? Also, what a “significant module of the PPI network” is needs to be clarified, and also what the software CytoHubba does.

Could the authors also please specify the versions of software used to predict the targets of miRNAs, and also, were all targets predicted for each gene used? Was some kind of cut-off value for confidence used?

In general, citations to all packages used need to be added, as at the moment only some of these have a reference.

Validity of the findings

Given that there are many details missing from the Methods section, it is difficult to establish how sound their results are. Specific concerns:

- It is mentioned that 514 thyroid cancer samples were analysed, but only 59 matched normals. How were the differentially expressed genes obtained then? I would think that only the tumour tissues with matched normal could be used for such comparison, but 514 tumour samples seems excessive. Can the authors clarify this point please?

- It is mentioned that 7/72 miRNAs were related to overall survival. How likely is it that this result is purely due to chance? Looking at the Kaplan-Meier survival curves, these look very similar (for example, b and f could be the same graph, at least by eye). It seems to me that it is the same two groups of patients (or two very similar groups) in all graphs. It could very well be that any these 7 miRNAs distinguishes these two groups of patients (although their effects are very possibly highly correlated), but it would be good if this analysis could be replicated in another dataset – is there any other available where this could be done? Reviewing the literature, it does seem that miR-146b has also been found upregulated in thyroid tumours vs normal (though only in PTC), but I have not found evidence for the other miRNAs. There are other studies exploring this issue, which the authors do mention in the Discussion – but only mentioning those results that agree with theirs. Could the authors add a note speculating why their results might be different from these, especially as it seems that miR-221 and miR-222 have been described previously but do not come up in the authors’ analyses?

- It is stated that 6 target genes regulated by 3 miRNAs were found in a list of 1,766 genes. How likely is it to find this result by chance? If the authors perform a randomisation experiment in which they iteratively select 1,766 genes from the genome, how often do they find 6 or more target genes of the 7 chosen miRNAs?

- It is difficult for me to understand whether the gene ontology analyses mean anything at all. It seems like a long list of processes and pathways – nearly any list inputted into DAVID will give a result. How is this result meaningful?

- Figure 3 is hard to interpret – all comparisons showed there seem to be significant (they have an asterisk above the two groups in all cases). Are they only showing the significant comparisons among all possible comparisons? And if so, did they correct for multitesting? What test was used to compare the groups? The effect sizes seem to be vanishingly small - Significance may arise from the number of individuals in each group but effects may not be meaningful in a clinical setting. Could the authors discuss a bit about the utility of this result as a prognostic factor?

Additional comments

In general and to summarise, I believe the most important points to address before this study can be accepted for publication are:

- Addition of detailed description of Methods
- Perform randomizations / multiple testing corrections where needed to control for false positives
- Discuss results that agree with the authors’ results, as well as those that don’t

---

## Round 0.2 · Minor Revisions

Thank you for addressing the comments from both reviewers. There are a few more minor changes required (see comments from reviewer 2) prior to being able to accept this paper. I am looking forward to seeing a revised version of the manuscript.

·

Basic reporting

This manuscript does a good job on revision. I am happy with the improved figures and languages. This work supply useful candidate genes and theoretical basis of mechanisms for thyroid cancer research.

Experimental design

I was pleased to see more details were added to the methods part. The experiment was well designed.

Validity of the findings

No comment.

Additional comments

Overall, the authors did a good job on revision. All my questions were well addressed. I agree to accept this manuscript.

·

Basic reporting

The authors have added substantially more detail to their methods, I am thankful for that as it is much easier to follow the performed analyses now. They have also added more references, discussed previous results, and added a couple of figures to clarify their methods.

I do believe the study is much more complete now and is suitable for publication.

Experimental design

The research question is well defined. I believe that the authors have followed logical steps to reach their conclusions. Methods now have sufficient detail for replication.

Some minor comments:

Can a sentence clarifying that a network interaction analysis was performed be added to the abstract, before talking about the top 10 hub genes? That would help make the abstract more understandable.

In line 148, should it say FDR < 0.05?

In line 174, would it be more understandable if it were worded as “A total of 573 samples with microRNA sequencing data and 573 samples with mRNA sequencing data were analyzed….”?

In line 176, it says that “RNA sequencing data included 510 thyroid cancer samples and 58 matched normal samples”, but this sums to 568 whereas the total number of samples analysed is stated as 573. What are the other 5?

Validity of the findings

This comment may stem from my limited experience with Kaplan-Meier survival curves, but I would suspect that the miRNAs and LPAR5 mRNA that are associated with survival are prognostic factors but not completely independent from each other: e.g. the survival curves for each of these look very similar. However, if these have been all evaluated together (i.e. each of these RNAs has an effect even after controlling for all others), then I am happy with the conclusions.

For this revision, the authors performed a replication of their results with additional available datasets, a very welcome addition to their study. This additional analysis has a large overlap with their own results, which lends credibility to their methods and conclusion.

Additional comments

I thank the authors for their answers and for adding details to their text and performing the additional suggested analyses.

---

## Round 0.3 · accepted · Accept

Thank you for addressing all of the reviewers' comments. Your manuscript is now accepted for publication.

#